# Seroprevalence and geospatial epidemiology of yaws: Evidence from Ghana

Abigail Agbanyo[1,2], Michael Ntiamoah Oppong[1], Ruth Dede Tuwor[1], Shadrach Mintah[3], Victor Yaw Morgan[1], Clement Tettey[4], Joseph Azabire[5], Owusu Boakye Yiadom[6], Dennis Odai Laryea[7], Alex Owusu-Ofori[2,8], Yaw Ampem Amoako [1,2,8]*, Richard Odame Phillips[1,2,8]

1 Kumasi Centre for Collaborative Research in Tropical Medicine, Kwame Nkrumah University of Science and Technology, Kumasi, Ghana, 2 School of Medical Sciences, Kwame Nkrumah University of Science and Technology, Kumasi, Ghana, 3 School of Public Health, Kwame Nkrumah University of Science and Technology, Kumasi, Ghana, 4 Regional Health Directorate, Western, Ghana Health Service, Sekondi, Ghana, 5 Wassa Amenfi East District Health Directorate, Ghana Health Service, Wassa Akropong, Ghana, 6 Aowin District Health Directorate, Ghana Health Service (GHS) Enchi, Ghana, 7 Disease Surveillance Department, Ghana Health Service, Accra, Ghana, 8 Komfo Anokye Teaching Hospital, Kumasi, Ghana

* yamoako2002@yahoo.co.uk

## Abstract

### Background

Yaws, a neglected tropical disease caused by *Treponema pallidum* subsp. *pertenue* remains a public health challenge in endemic regions. Although the World Health Organization (WHO) has earmarked yaws for eradication by the year 2030, there is a dearth of accurate epidemiological data to facilitate eradication efforts. The WHO recommends the use of seroprevalence surveys and geospatial analysis to guide planned interventions.

### Methodology and findings

We conducted a cross-sectional study in Wassa Amenfi East and Aowin districts in Ghana, clinically screening 11,505 school children for yaws. Treponemal Rapid Diagnostic Test (RDTs) detected 117 individuals, and the Dual Path Platform (DPP) confirmed 73 cases, giving an overall prevalence of 0.63%; and 3.85% for Aowin district compared to 0.31% for the Wassa Amenfi East district. Prevalence among RDT-tested was 7.79%, (34.54% from Aowin and 4.42% from Wassa Amenfi East) and DPP seroprevalence was 62.39% (Aowin, 70.69% and Wassa Amenfi East, 54.24%). A prevalence rate of 0.02% for latent infection was recorded in Wassa Amenfi East. Spatial analysis employing multiple mapping techniques including spatial autocorrelation analysis (Moran's I), kriging, nearest neighbour analysis, and kernel density estimation using data from the surveyed communities indicated significantly clustered hotspots in Aowin's central and Wassa Amenfi East's southeastern part. Kriging interpolation with barriers and Empirical Bayesian kriging revealed consistent spatial trends in unsurveyed communities in the study area.

**Data availability statement:** All relevant data are within the manuscript and its Supporting Information files.

**Funding:** The author(s) received no specific funding for this work.

**Competing interests:** The authors have declared that no competing interests exist.

## Conclusion

This study found an overall yaws prevalence of 0.63% but with significantly clustered hotspots in some parts of the districts. The findings from this study highlight the importance of serosurveys and geospatial modeling in generating critical data to guide surveillance, education, and mass drug administration (MDA) efforts in endemic regions to support the WHO's goal of eradicating yaws by 2030.

### Authors summary

Yaws is a skin disease caused by the bacterium *Treponema pallidum* subsp. *pertenue* and mostly affects people in poor, tropical areas. It is transmitted from one person to the other through direct skin contact with an affected person. Yaws is still a major health problem in places where it is common, and although children are the most affected, all age groups can be affected. The World Health Organization (WHO) wants to completely wipe out the disease by the year 2030. To do this, we need to know exactly how many people have the disease and where they are. This means using the right diagnostic tools in communities and schools to actively look for both those who have the disease and are showing signs, and those who have the disease but are without signs. However, many countries only report cases based on clinical signs without proper testing. To properly fight and end yaws, we need more accurate information. The WHO recommends using blood tests (seroprevalence surveys) and maps (geospatial analysis) to understand how common the disease is in different areas. This will help endemic communities plan better and track their progress over time. In our study, we found an overall yaws prevalence of 0.63%. Seroprevalence among RDT-tested was 7.79% and DPP seroprevalence was 62.39% (Aowin, 70.69% and Wassa Amenfi East, 54.24%). A prevalence rate of 0.02% for latent infection was recorded in Wassa Amenfi East. Spatial analysis employing multiple mapping techniques using data from the surveyed communities revealed significantly clustered hotspots in Aowin's central and Wassa Amenfi East's southeastern part. We found consistent spatial trends in unsurveyed communities in the study area.

Our findings show the importance of serosurveys and geospatial modeling in generating critical data to guide surveillance, education, and mass drug administration (MDA) efforts in endemic regions to support the WHO's goal of eradicating yaws by 2030.

## Introduction

Yaws is a non-venereal treponemal infection caused by *Treponema pallidum* subsp. *pertenue.* The disease primarily occurs in warm and humid regions [1] and is mainly associated with economically and resource-disadvantaged communities in tropical and subtropical areas of the world [2].

Transmission occurs through direct skin-to-skin contact with infected lesions, causing lesions that affect the skin, cartilage, and bones after an incubation period of 9–90 days [3]. Children under the age of 15 years account for about 75–80% of all cases reported globally and serve as the main reservoir of the disease [4,5]. The initial clinical manifestations, such as papules, macules, ulcers, and plaques, may progress to more disfiguring advanced stages if left untreated [6,7].

Sixteen (16) countries are currently classified as endemic for yaws, while the situation in about 82 previously identified endemic countries, regions, and territories remains unclear [8]. Over the years, Papua New Guinea, Ghana, the Solomon Islands, and more recently, Indonesia have accounted for the highest numbers of suspected yaws cases globally [9]. The Dual Path Platform Syphilis Screen and Confirm assay (Chembio Diagnostics, USA) is recommended by the WHO for confirming yaws cases, as it detects both treponemal and non-treponemal antibodies, allowing differentiation between past treated infections and active disease [4]. However, diagnostic confirmation of cases remains a significant challenge, likely due to the high cost and limited availability of diagnostic tests in many endemic regions [10].

Ghana remains endemic for yaws and appears to have the highest number of cases in the WHO African region [11,12]. In 2012, yaws was reported in 160 out of 170 (94.1%) health districts in Ghana [13,14].

Although a large-scale campaign in the 1950s significantly reduced the number of yaws cases globally, the lack of active surveillance led to a resurgence, making yaws a pressing public health concern today [15]. The WHO Morges Eradication Strategy, which aimed to eliminate the disease by 2020 [1,16] involved two mass drug administration (MDA) approaches: total community treatment (TCT) and total targeted treatment (TTT), both using a single oral dose of azithromycin (30 mg per kg of body weight). Although the original target was missed, the goal has now been extended to 2030 [17]. Despite the optimism about achieving this eradication target, which relies largely on accurate surveillance data, clinical and epidemiological data remain limited. Many countries report only clinically suspected cases without laboratory confirmation [9,12] and there is inadequate data to guide control programmes in many countries. To meet this eradication target, there is a need for accurate epidemiological data, including seroprevalence and spatial distribution, to guide targeted interventions and monitor progress towards eradication.

Ghana's policy to eliminate yaws has consistently aligned with the WHO's eradication roadmap. The WHO recommends serosurveys and geostatistical approaches to identify and understand local variations of yaws to facilitate effective targeted interventions [18]. We aimed to establish the seroprevalence of yaws infection, map and analyze the spatial patterns of yaws in two health districts in Ghana to understand the local epidemiology, and to inform effective intervention strategies.

## Methods

### Ethical approval and informed consent

The study received approval from the Committee on Human Research, Publications, and Ethics (CHRPE) of the Kwame Nkrumah University of Science and Technology (KNUST) under reference numbers CHRPE/AP/361/24 and CHRPE/AP/540/24. Written informed consent was provided by the parents or guardians of all participants before screening during the case search activities. The content of the participant information leaflet was explained to all participants and their legal representatives with the assistance of a teacher fluent in the local dialect, mostly Twi, where needed. Furthermore, assents were obtained from children aged 12–17 years with DPP positive results before treatment. This study was conducted in accordance with the ethical principles outlined in the Declaration of Helsinki [19].

### Study setting

The study was conducted in two districts, Wassa Amenfi East district in the Western region and the Aowin district of the Western North region of Ghana. These two districts are located in the southwestern part of Ghana, near the country's border with Cote d'Ivoire. With an estimated population of 179,696, the Wassa Amenfi East district has Wassa Akropong as the capital and lies between latitudes 5°30' N and 6°15' N, and longitudes 1°45' W and 2°11' W. The district is divided into

7 subdistricts [20] and can boast of public health facilities, including 1 hospital, 6 health centres, and 41 Community-based Health Planning and Services (CHPS) facilities [21]. Some communities are located within the forest zones near the existing district boundary lines; however, these are all considered part of the district [21]. Aowin has Enchi as its capital and a population of about 129,721 individuals. Located between latitudes 5°25' N and 6°14' N, and longitudes 2°30' W and 3°05' W, the district is divided into 9 health subdistricts and has 1 hospital, 11 health centres, 4 clinics, and 27 CHPS facilities [22,23].

The districts are largely rural, with approximately 77% and 87% of the populations in Wassa Amenfi East and Aowin, respectively, residing in rural communities, and mostly engaged in farming and small-scale mining activities. In both districts, children under 15 years form a substantial proportion (approximately 40%) of the population [24].

These two districts were considered for the study because they had inadequate published epidemiological data on yaws before our survey. Our group previously published a study that included cases of yaws from the Wassa Amenfi East district [25]; however, there are no readily available prevalence studies in the Aowin district. At the time of this survey, no recorded mass treatment interventions had been rolled out in the Aowin district.

## Study design and participant recruitment

This was a cross-sectional study conducted between May and December 2024. In each district, the choice of schools to visit was guided by the District Health Management Team (DHMT), relying on historical reporting trends for yaws within the area. In Wassa Amenfi East, a total of 33 basic schools in 29 communities were visited, while 4 basic schools in 4 communities were visited in Aowin district (Fig 1). Based on the known epidemiology of yaws, children in the basic schools who typically fall in the key transmission age [4] were included in the study, along with children aged 1–5 years who, although typically not enrolled in basic schools, may be found in kindergarten schools and are key for ongoing disease transmission [18]. Due to the focal endemicity and uneven distribution of yaws as a disease, no formal sample size calculation was done, and the sample size was based on the convenience of sampling in the selected schools.

## Community sensitization and case search

Activities began with community entry, sensitization, and social mobilization in each community. Key traditional and school authorities, such as village chiefs, opinion leaders, local administrative leaders, teachers, and parents, were engaged to introduce the study and seek permission to undertake activities within the schools. Announcements about the study were made through the local community information systems in each community as a means of giving information to community members. Community Health Workers (CHWs) used culturally sensitive communication in local languages to address concerns and dispel misconceptions.

The case search activities within the selected schools were conducted in partnership with the Disease Control unit at the District Health Directorate and the District Education Directorate. The district health staff received refresher training on the identification of yaws as outlined in the WHO's Yaws Recognition Booklet for Communities [26] by two Consultant Physicians (YAA and ROP) with extensive expertise in the clinical and public health aspects of neglected tropical diseases of the skin (skin NTDs).

## Skin examination and serological testing

In each school, a comprehensive body screening was conducted for all children present during the visit. As part of routine procedures, children who were absent from school but were reported to have lesions and were brought in by parents or legal guardians were also examined. Additionally, children who had previously experienced untreated, resolved, or clinically similar lesions before our visit were screened serologically. The screening and clinical diagnosis were carried out by clinicians, researchers, and Disease Control Officers with expertise in skin NTDs.

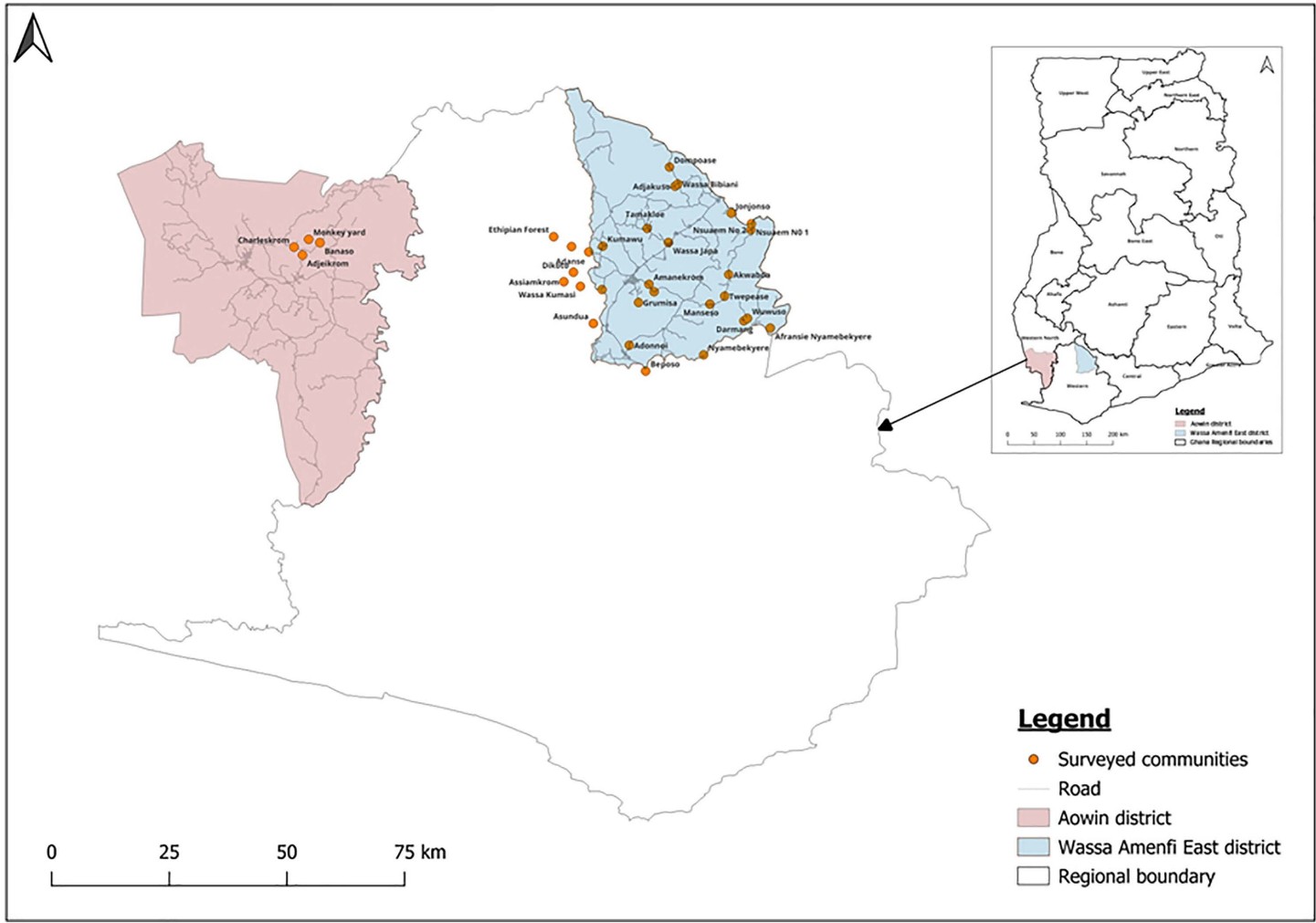

**Fig 1. Map of Ghana showing the study sites and the geographical distribution of surveyed communities in the two districts.** Map was generated using ArcGIS 10.7.1 (Esri Inc., Redlands, California, USA). The shapefiles for Ghana and the various regions obtained from OpenStreetMap (https://www.openstreetmap.org/copyright, CC BY-SA 2.0) were utilized as data sources for plotting the map. Map data from © OpenStreetMap https://www.openstreetmap.org/copyright.

The skin examination involved assessing exposed areas, including the feet, legs up to the knees, hands up to the upper arms, neck, face, and scalp. Students remained in their school uniforms, which mostly consisted of above-knee shorts and above-elbow shirts (for males) or dresses (for females). Shoes and socks were removed before the examination. In each school, examinations were conducted in a dedicated classroom accessible only to the study team and a chaperone to ensure privacy. The examination primarily included the entire upper half of the body and exposed areas of the lower body. Unexposed areas were only examined if specifically reported by the individual. The breasts and genitals were excluded unless requested by participants, in which case they were assessed in a separate, private examination area. For females, appropriate measures were taken to ensure that the breasts remained covered. As far as was possible, examinations were conducted by gender-appropriate personnel.

Individuals with suspected yaws lesions were tested using the rapid point-of-care treponemal test (SD Bioline Syphilis 3.0 RDT kit, Standard Diagnostics Inc., Suwon, South Korea) in accordance with the manufacturer's instructions. Further,

all individuals with positive syphilis RDTs were confirmed using a rapid treponemal and non-treponemal test (DPP Syphilis Screen and Confirm Assay, Chembio Diagnostic Systems, Medford, NY, USA).

### Definition of terms

A suspected case of yaws was defined as an individual presenting with clinical signs consistent with primary and secondary yaws, including skin papilloma, solitary or multiple atraumatic painless skin ulcerations with typically rolled edges, squamous macules, bone swelling, and palmar or plantar lesions [7,25].

A probable case of yaws was defined as a suspected yaws case with a positive result on the rapid point-of-care treponemal test (SD Bioline Syphilis 3.0 RDT kit, Standard Diagnostics Inc., Suwon, South Korea) [25].

A confirmed yaws case was defined as a probable case of yaws with positive results on both treponemal and non-treponemal tests using the DPP test, read 15 minutes after the test, according to the manufacturer's instructions [27].

An active yaws case was defined as a symptomatic case with primary or secondary yaws lesions with positive results on both treponemal and non-treponemal tests using the DPP [28].

A latent yaws case was defined as an asymptomatic individual with positive results on both treponemal and non-treponemal tests using the DPP [28].

### Treatment

Children with skin lesions not consistent with yaws were referred to the local health facility for diagnosis and management. All confirmed DPP cases received oral azithromycin at a dose of 30mg/kg up to a maximum of 2g following the dosage guide recommended by the WHO [29], administered by trained community health nurses. Participants with ulcers were provided with simple gauze dressings to protect against trauma and keep lesions clean.

### Data management

Using a standardized WHO Skin disease reporting and recording form [30], demographic and clinical data were obtained from all screened participants. Data was entered and managed with Microsoft Excel version 2019 (Microsoft Corp., Redmond, WA, USA). Global Positioning Systems (GPS) coordinates of all schools visited were recorded using the My GPS Essentials App (GPS Essentials, Seattle, CA, USA).

### Statistical analysis

Data analysis for this yaws study was conducted using GraphPad Prism version 10.5.0 (GraphPad Software, San Diego, CA, USA). Demographic characteristics, including age and gender, were summarized using medians and interquartile ranges (IQR) for continuous variables, and proportions and percentages for categorical variables, with stratification based on the participants' district of residence.

Prevalence of yaws in each district was calculated based on the DPP positive cases, expressed as a percentage of the population clinically screened for each district.

Seroprevalence based on RDT was calculated based on the RDT-positive cases expressed as a percentage of the population serologically screened with the RDT test in each district to detect the proportion of the population having detectable antibodies to the treponemal antigen, indicating both past exposure and active cases.

Seroprevalence based on DPP was similarly determined based on the confirmed DPP-positive cases expressed as a percentage of the population serologically tested with the DPP to detect only active cases.

Adjusted prevalence ratios were calculated using RStudio version 2023.12.0.369 (RStudio: Integrated Development Environment for R. Posit Software, PBC, Boston, MA, USA) with the survey package to account for differences in school population sizes, providing a more reliable measure for spatial comparisons.

## Mapping and geostatistical modelling

Geospatial analysis of yaws was conducted using adjusted prevalence data from the surveyed communities to assess spatial patterns and predict disease distribution within the study districts, employing multiple mapping techniques including spatial autocorrelation analysis (Moran's I), Kriging, Nearest Neighbour Analysis, and Kernel Density Estimation (KDE) for the identification of high-risk areas.

Global Moran's I spatial autocorrelation technique, which is essential before conducting any spatial clustering analysis [31], was used to determine if yaws prevalence pattern in the study area was clustered, dispersed, or random with a z-score value and a significant p value (p < 0.05) using prevalence data from the survey. Once there is clustering or randomness, Local Moran's I [Local Indicator of Spatial Association (LISA)] [32,33] helps determine where local clustering is observed. A cluster map was created to visually identify statistically significant groupings of areas with aggregation of yaws cases in the study area. Surveyed areas were classified as High-High (HH), Low-Low (LL) clusters, High-Low (HL), or Low-High (LH) outliers, with significance at p < 0.05. HH clusters indicated areas where high prevalence was surrounded by other high prevalence areas, LL cluster represented areas with low prevalence adjacent to other low-prevalence areas, HL outlier, denoted high prevalence areas bordered by low prevalence areas, and the LH outlier represented areas with low prevalence adjacent to high prevalence areas. Areas of no significance were also determined based on the prevalence recorded in the survey [32].

A kernel density map of the intensity of yaws prevalence across the districts was created to visualize the continuous density of yaws cases, highlighting areas of high density using prevalence data from the survey and community proximity from geocoordinates collected for the communities [34]

After identifying the spatial autocorrelation and density of yaws in the study area, hotspot analysis was used to determine high-risk areas for targeted intervention. Hotspot maps using Getis-Ord Gi statistics [35–37] was performed to confirm significant clusters and identify significant hotspots (high prevalence area) and coldspots (low prevalence area), to align with community proximity.

An area was considered a yaws hotspot when the area with high prevalence is surrounded by a cluster of high yaws cases, and an area was considered a coldspot when the area with low yaws is surrounded by a cluster of low yaws cases. This analysis returns an output feature with a p-value, Z-score, and confidence interval bin field (Gi-Bin) with a fixed distance band. The Gi-Bin includes features such as 90%, 95%, and 99% confidence levels, indicating hotspots, while cold spots were denoted with the same confidence intervals, denoting the strength of evidence for clustering. Features of the Gi-Bin values near 0 were considered non-significant areas, indicating no substantial clustering [35].

Finally, a prediction map for unsurveyed areas in the study site was developed through spatial interpolation methods. Both Kernel interpolation with barriers using boundaries river bodies in the district and Empirical Bayesian Kriging interpolation rely on spherical semi-variogram modeling and were implemented to estimate prevalence rates of unsurveyed areas expressed as percentages across the unsurveyed districts based on prevalence data from surveyed communities [38]. Maps and spatial analysis were generated using ArcGIS 10.7.1 (Esri Inc., Redlands, California, USA). Publicly available shapefiles from OpenStreetMap (www.openstreetmap.org) were used for base mapping.

## Results

### Demographic characteristics

A total of 11505 children (Table 1 and Fig 2), with a median age of 10.8 years (IQR: 7.8-13.7), were screened from 37 basic schools across 33 communities within the study area. The participants were from Wassa Amenfi East district (n = 10439, representing 33 schools in 29 communities) and Aowin district (n = 1066, from 4 schools in 4 communities).

The overall cohort consisted of 51% males (n = 5830) and 49% females (n = 5675). The majority (5048/11505; 44%) of screened children were aged 11–15 years. The age distribution showed that 9% were 5 years or younger, 42% were between 6–10 years, 44% were between 11–15 years, and 5% were over 15 years.

Of the children examined clinically, 13% (1502/11505) either showed lesions suggestive of yaws or had previously had similar lesions that resolved without treatment. These cases were tested with the RDT. Of these, 1334 cases were from Wassa Amenfi East (comprising 1330 with clinically suspected yaws and 4 asymptomatic individuals reporting prior untreated similar lesions) and 168 were from Aowin districts. Overall, 117/1502 (7.8%) tested positive on the RDT: 59 from Wassa Amenfi East, including 2 of the 4 asymptomatic individuals, and 58 from Aowin. The median age of RDT-positive individuals was 12 years (IQR: 9–13). A total of 73/117 individuals, with a positive non-treponemal component on DPP testing, were confirmed to have yaws: 32 from Wassa Amenfi East and 41 from Aowin. The median age of those with confirmed yaws was 11 years (IQR: 8.3-13).

## Serological findings

Table 2 provides a summary of the prevalence and seroprevalence within the study area. The study recorded an overall prevalence using DPP of 0.63% (73/11505). For Aowin, the prevalence was 3.85% (41/1066) and 0.31% (32/10439) in Wassa Amenfi East district. Overall, males had a higher prevalence (0.96%; 56/5830) than females (0.30%; 17/5675). Further, the prevalence among males in Aowin was 5.60% (30/536) compared to 0.49% (26/5294) in Wassa Amenfi East. Children older than 10 years had the highest prevalence of 0.73% (41/5583) compared to the other age groups (0.18%; 2/1096 in those ≤5 years and 0.62% (30/4826) in those 6–10 years old. Active yaws prevalence was 0.62% (71/11501) in the study districts [3.85% (41/1066) in Aowin and 0.29%; (30/10435) in Wassa Amenfi East], while an overall 0.02% (2/11505) latent yaws infection was detected only in Wassa Amenfi East through serological testing.

Among participants with a positive RDT, the overall seroprevalence was 7.79% (117/1502). Aowin showed a higher rate (34.52%; 58/168) compared to Wassa Amenfi East (4.42%; 59/1334), indicating the presence of both past and active

**Table 1. Demographic characteristics of study participants.**

| | Wassa Amenfi East district | | | | Aowin district | | | | All | | | |
|---|---|---|---|---|---|---|---|---|---|---|---|---|
| | Total Screened (%) | RDT Tested n, (%) | RDT Positive n, (%) | DPP Positive n, (%) | Total Screened n, (%) | RDT Tested n, (%) | RDT Positive n, (%) | DPP Positive n, (%) | Total Screened n, (%) | RDT Tested n, (%) | RDT Positive n, (%) | DPP Positive n, (%) |
| N | 10439 | 1334 | 59 | 32 | 1066 | 168 | 58 | 41 | 11505 | 1502 | 117 | 73 |
| Median Age, years (IQR) | 10.9 (7.9-13.7) | 10 (8-13) | 12 (9.3-13) | 11 (8-13.3) | 9.6 (6.6-12.8) | 10 (7-13) | 12 (9-14) | 11 (8-13) | 10.8 (7.8-13.7) | 10 (8-13) | 12 (9-13) | 11 (8.3-13) |
| Age category (years) ≤5 | 885 (8) | 99 (7) | 0 (0) | 0 (0) | 211 (20) | 19 (11) | 2 (3) | 2 (5) | 1096 (9) | 118 (8) | 2 (2) | 2 (3) |
| 6-10 | 4376 (42) | 572 (43) | 19 (32) | 13 (41) | 450 (42) | 71 (42) | 21 (36) | 17 (41) | 4826 (42) | 643 (43) | 40 (34) | 30 (41) |
| 11-15 | 4662 (45) | 597 (45) | 36 (61) | 18 (56) | 386 (36) | 75 (45) | 32 (55) | 20 (49) | 5048 (44) | 672 (45) | 68 (58) | 37 (51) |
| >15 | 516 (5) | 66 (5) | 4 (7) | 1 (3) | 19 (2) | 3 (2) | 3 (5) | 2 (5) | 535 (5) | 69 (4) | 7 (6) | 4 (5) |
| Male (%) | 5294 (51) | 805 (60) | 44 (75) | 26 (81) | 536 (51) | 106 (63) | 41 (71) | 30 (73) | 5830 (51) | 911 (61) | 85 (73) | 56 (77) |
| Female (%) | 5145 (49) | 529 (40) | 15 (25) | 6 (19) | 530 (49) | 62 (37) | 17 (29) | 11 (27) | 5675 (49) | 591 (39) | 32 (27) | 17 (23) |
| Active yaws | 10435 (99.9) | 1330 (99.7) | 57 (97) | 30 (94) | 1066(100) | 168 (100) | 58 (100) | 41 (100) | 11501(99.9) | 1498 (99.7) | 115 (98) | 71 (97) |
| Latent yaws | 4 (0.04) | 4 (0.3) | 2 (3) | 2 (6) | 0 | 0 | 0 | 0 | 4 (0.04) | 4 (0.3) | 2 (2) | 2 (3) |

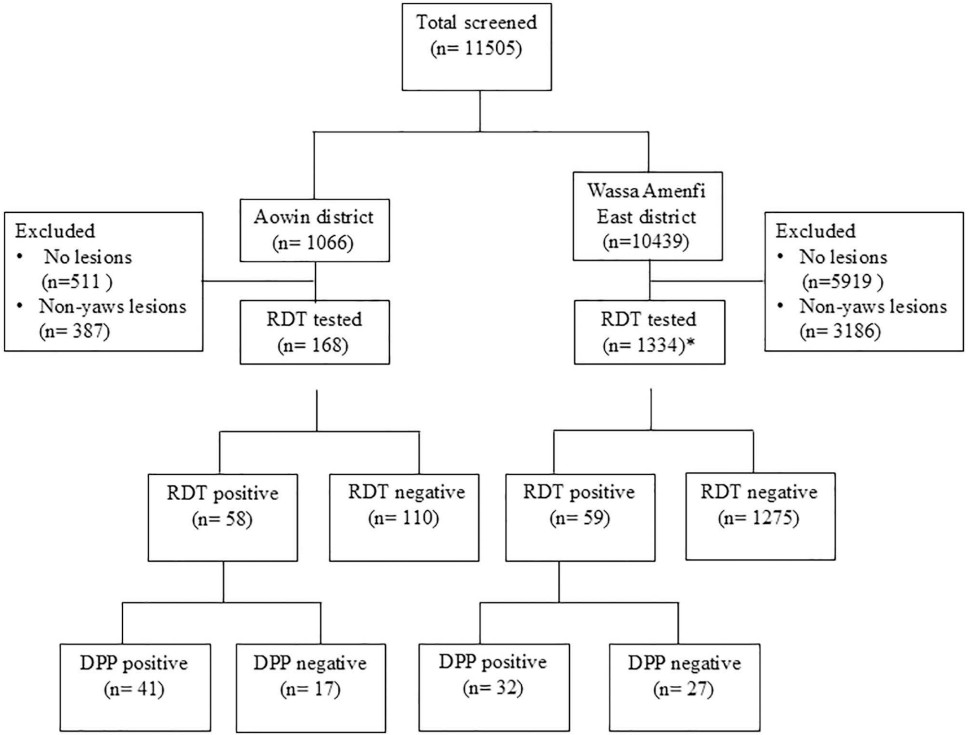

**Fig 2. Flow chart of study participants.** *The number of individuals tested with RDT in the Wassa Amenfi East district included persons with suspected yaws (n = 1330) and 4 asymptomatic persons who were reported to previously have clinical lesions but received no treatment (2 out of these 4 individuals had a positive RDT).

**Table 2. Prevalence and seroprevalence of yaws across study districts.**

| | Wassa Amenfi East | | | Aowin | | | All | | |
|---|---|---|---|---|---|---|---|---|---|
| | Prevalence % | Seroprevalence, RDT % | Seroprevalence, DPP % | Prevalence % | Seroprevalence, RDT % | Seroprevalence, DPP % | Prevalence % | Seroprevalence, RDT % | Seroprevalence, DPP % |
| Overall | 0.31 | 4.42 | 54.24 | 3.85 | 34.52 | 70.69 | 0.63 | 7.79 | 62.39 |
| Age category (years) | | | | | | | | | |
| ≤5 | 0 | 0 | 0 | 0.95 | 10.53 | 100 | 0.18 | 1.69 | 100 |
| 6-10 | 0.30 | 3.32 | 68.42 | 3.78 | 29.58 | 80.95 | 0.62 | 6.22 | 75.00 |
| 11-15 | 0.39 | 6.03 | 50.00 | 5.18 | 42.67 | 62.50 | 0.73 | 10.12 | 54.41 |
| >15 | 0.19 | 6.06 | 25.00 | 10.53 | 100 | 66.67 | 0.75 | 10.14 | 57.14 |
| Gender | | | | | | | | | |
| Male (%) | 0.49 | 5.47 | 59.09 | 5.60 | 38.68 | 73.17 | 0.96 | 0.96 | 65.88 |
| Female (%) | 0.12 | 2.84 | 40.00 | 2.08 | 27.42 | 64.71 | 0.30 | 0.30 | 53.13 |
| | **Prevalence** | | | **Prevalence** | | | **Prevalence** | | |
| Active | 0.29 | | | 3.85 | | | 0.62 | | |
| Latent | 0.02 | | | 0 | | | 0.02 | | |

infections. As expected, seroprevalence based on RDT positivity increased with age. Participants up to 5 years old had an RDT seroprevalence of 1.69%; 2/118 (with Aowin at 10.53%; 2/19 and Wassa Amenfi East at 0%).

Seventy-three participants had a positive DPP result, giving a seroprevalence of 62.39% (73/117); Aowin district recorded a higher rate (70.69%; 41/58) compared to Wassa Amenfi East district (54.24%; 32/59). There was an overall 100% (2/2) rate of DPP seroprevalence in individuals up to 5 years, reported from Aowin district (Table 2).

## Characteristics of DPP confirmed yaws cases

Table 3 summarizes the clinical characteristics of the 73 individuals with confirmed yaws. Ulcers were the most common manifestation, seen in 48 participants (65.8%; 48/73). The primary stage of the disease was seen in 45 participants (61.6%; 45/73), including 41 (56.2%) single ulcers (Aowin, 73.2% and Wassa Amenfi East, 34.4%) and 5.5% papillomas (Aowin, 2.4% and Wassa Amenfi East, 9.4%). Squamous macules were noted in 12 participants (16.4%), comprising 4 (9.8%) in Aowin and 8 (25.0%) in Wassa Amenfi East. Palmar and plantar lesions were identified in 7 participants (9.6%), with 3 (7.3%) in Aowin and 4 (12.5%) in Wassa Amenfi East, while asymptomatic cases were rare, occurring in 2 participants (2.7%), both in Wassa Amenfi East.

The location of lesions varied between districts. The leg was the most commonly affected site, with 34/73 cases (46.6%), including 21 (51.2%) in Aowin and 13 (40.6%) in Wassa Amenfi East. The sole of the foot was involved in 11 cases (15.1%), comprising 4 (9.8%) in Aowin and 7 (21.9%) in Wassa Amenfi East. The knee accounted for 8 cases (11.0%), mainly in Aowin with 7 (17.1%), compared to 3 (3.1%) in Wassa Amenfi East. The ankle was affected in 5 cases (6.8%), all in Aowin, while the elbow was involved in 3 cases (4.1%), also only in Aowin. Lesions on the arm, hand, face,

**Table 3. Clinical characteristics of DPP confirmed yaws cases.**

| Characteristic | All n=73 (%) | Aowin district n=41 (%) | Wassa Amenfi East district n=32 (%) |
|---|---|---|---|
| **Clinical characteristics** | | | |
| **Primary stage, (n=45)** | | | |
| Single ulcer | 41 (56.2) | 30 (73.2) | 11 (34.4) |
| Papilloma | 4 (5.5) | 1 (2.4) | 3 (9.4) |
| **Secondary stage, (n=26)** | | | |
| Disseminated ulcer | 7 (9.6) | 3 (7.3) | 4 (12.5) |
| Squamous macules | 12 (16.4) | 4 (9.8) | 8 (25.0) |
| Palmar and plantar | 7 (9.6) | 3 (7.3) | 4 (12.5) |
| Bone swelling | 0 (0.0) | 0 (0.0) | 0 (0.0) |
| **Latent stage, (n=2)** | | | |
| Asymptomatic | 2 (2.7) | 0 (0.0) | 2 (6.3) |
| **Location of lesions** | | | |
| Leg | 34 (46.6) | 21 (51.2) | 13 (40.6) |
| Ankle | 5 (6.8) | 5 (12.2) | 0 (0.0) |
| Sole of foot | 11 (15.1) | 4 (9.8) | 7 (21.9) |
| Arm | 4 (5.5) | 0 (0.0) | 4 (12.5) |
| Hand | 2 (2.7) | 0 (0.0) | 2 (6.3) |
| Face | 3 (4.1) | 0 (0.0) | 3 (9.4) |
| Head | 1 (1.4) | 0 (0.0) | 1 (3.1) |
| Leg, back, stomach, and arms | 1 (1.4) | 1 (2.4) | 0 (0.0) |
| Elbow | 3 (4.1) | 3 (7.3) | 0 (0.0) |
| Knee | 8 (11.0) | 7 (17.1) | 1 (3.1) |
| Hand and heel | 1 (1.4) | 0 (0.0) | 1 (3.1) |

and head were less frequent, with 4 (5.5%), 2 (2.7%), 3 (4.1%), and 1 (1.4%) case respectively in Wassa Amenfi East, and none reported in Aowin. One case in Aowin had lesions on the leg, back, stomach, and arms, while two cases in Wassa Amenfi East involved the hand and heel. These results show diverse clinical and anatomical presentations of yaws, with distinct district-specific differences in lesion types and locations (Table 3 and Fig 3).

## Mapping

Spatial analysis of yaws prevalence among 11,505 children screened across 37 communities in Wassa Amenfi East and Aowin districts revealed distinct patterns. Global Moran's I autocorrelation analysis (S1 File) identified significant spatial clustering in the study area, confirming that yaws is not randomly distributed (p < 0.01, z score, 4.5). To understand location specific spatial patterns, LISA (p < 0.01, z score 5.4) (S2 File) was used and cluster maps generated with High-High clusters in Aowin's central-western region and Low-High outliers in Wassa Amenfi East's southeastern areas, indicating concentrated and sparse prevalence, respectively, as visualized by a cluster map (Fig 4).

Kernel density estimation (Fig 5) showed a higher density of DPP-confirmed cases in Aowin, reflecting its 3.85% prevalence, compared to a lower density in Wassa Amenfi East (0.31% prevalence).

Hotspot analysis using Getis-Ord Gi (Fig 6) confirmed a significant hotspot in Aowin (4.03% prevalence, p < 0.05) at 99% Confidence and a hotspot in Wassa Amenfi East (0.29% prevalence, p < 0.05) at 90% Confidence, aligning with seroprevalence differences across the districts. Non-significant areas were identified in the study areas, with no cold spots identified. The average nearest neighbour summary (S3 File) reported a nearest neighbour ratio of 0.65 (p < 0.01), suggesting a cluster pattern with an average distance of 3.6km between cases, significantly less than the expected 5.5km for a random distribution.

The interpolation analysis of yaws prevalence in the unsurveyed study area was conducted with both Kernel interpolation with Barrier (Fig 7A) and Empirical Bayesian kriging (Fig 7B), revealing consistent spatial trends. District boundaries and river bodies in the district were used as barriers and estimated prevalences predicted elevated prevalence in

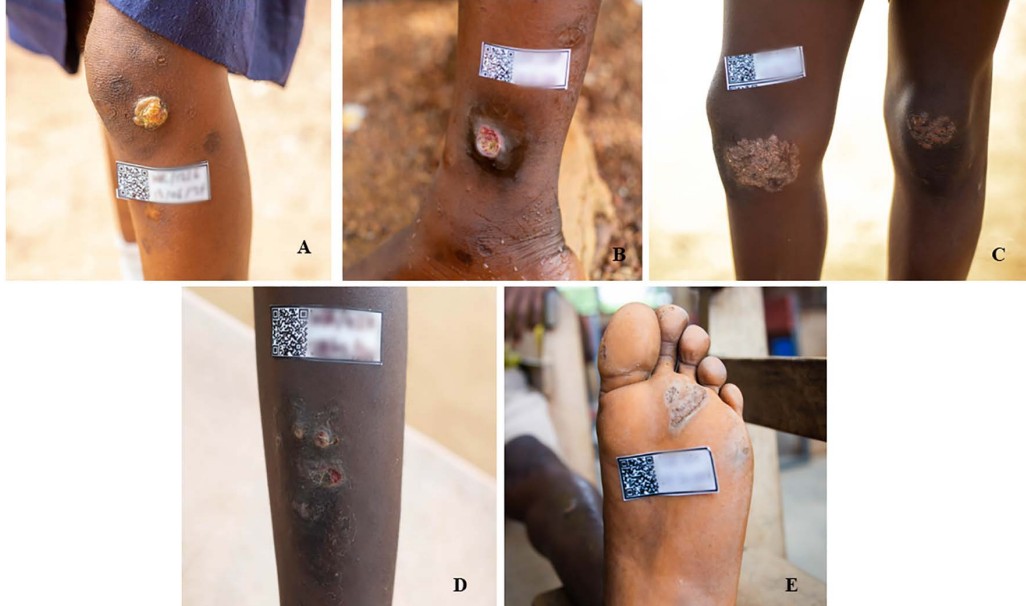

**Fig 3. Clinical lesions of DPP positive yaws cases.** Papilloma **(A)**, single ulcer **(B)**, squamous macules **C)**, disseminated ulcer **(D)**, plantar **(E)**.

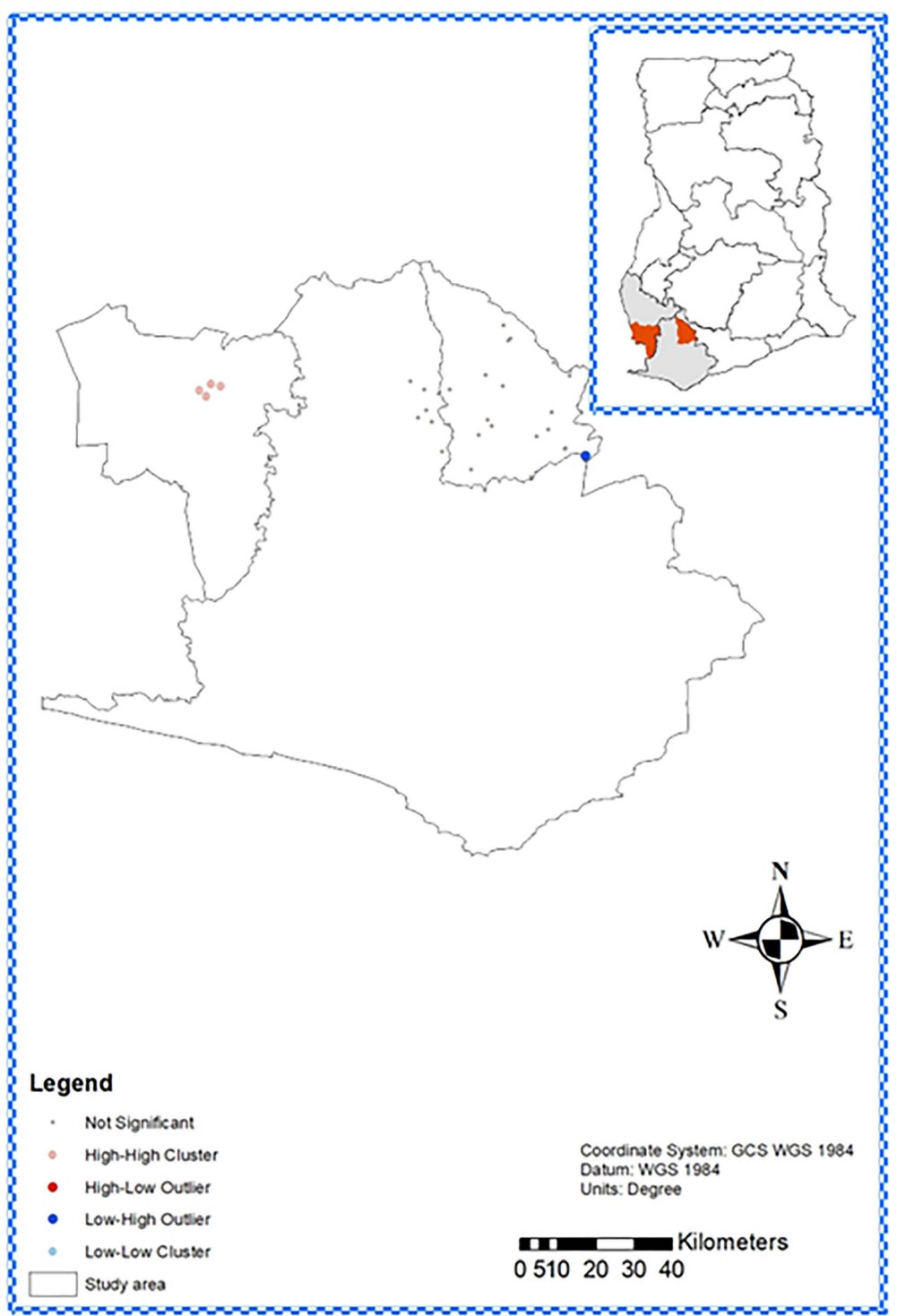

**Fig 4. Local clustering of yaws prevalence in Wassa Amenfi East and Aowin Districts, Ghana.** Map was generated using ArcGIS 10.7.1 (Esri Inc., Redlands, California, USA). The shapefiles for Ghana and the various regions obtained from OpenStreetMap (https://www.openstreetmap.org/copyright, CC BY-SA 2.0) were utilized as data sources for plotting the map. Map data from © OpenStreetMap. https://www.openstreetmap.org/copyright. Not significant -areas with no significant clusters; High-High (HH) cluster- areas where high prevalence was surrounded by other high prevalence areas; Low-low (LL) cluster- areas with low prevalence adjacent to other low-prevalence areas, High-Low (HL) outlier- high prevalence areas bounded by low prevalence areas; and the Low-High (LH) outlier- areas with low prevalence adjacent to high prevalence areas. Areas of no significance were also determined based on the prevalence recorded in the survey.

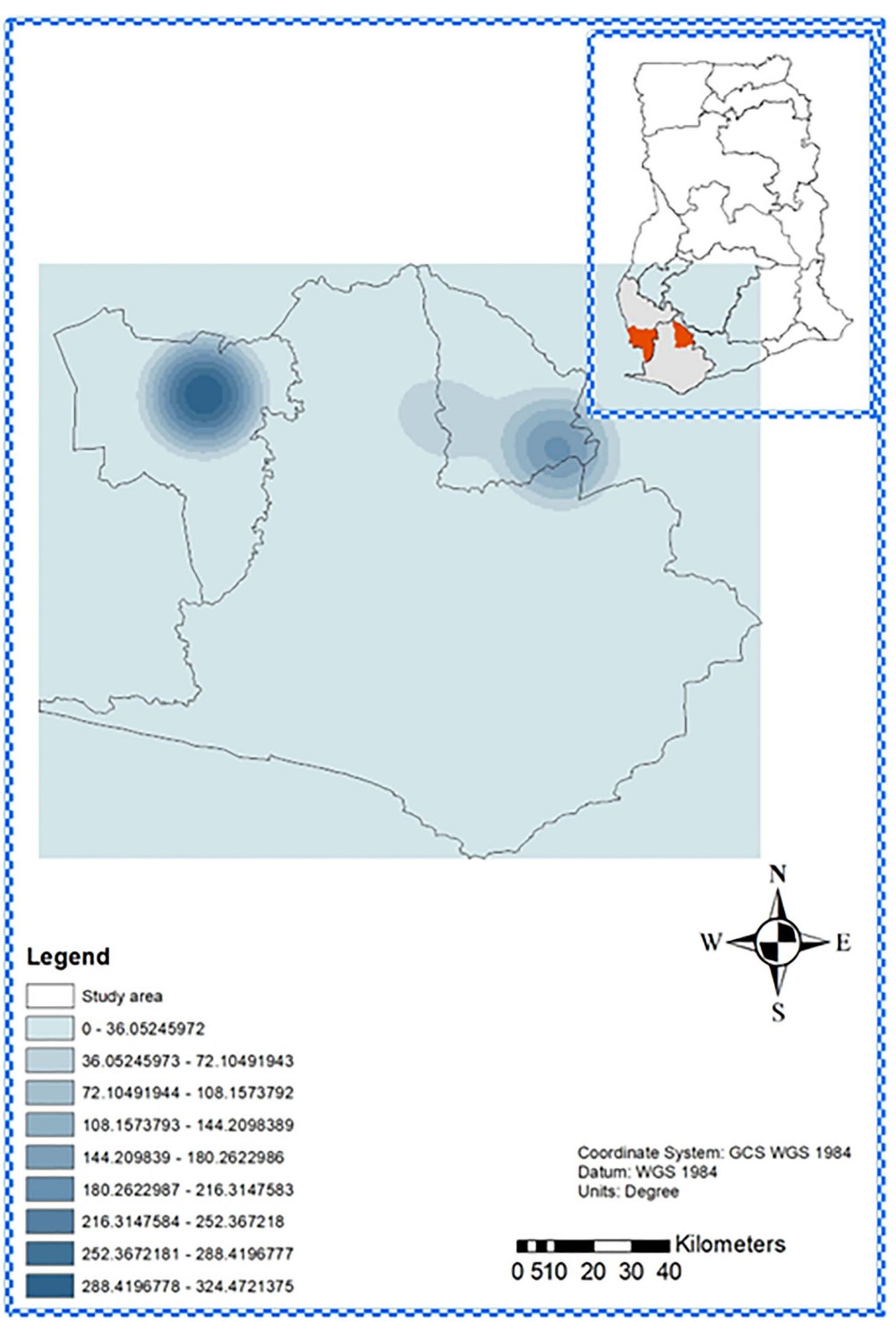

**Fig 5. Kernel Density of yaws cases in Wassa Amenfi East and Aowin Districts, Ghana.** Map was generated using ArcGIS 10.7.1 (Esri Inc., Redlands, California, USA). The shapefiles for Ghana and the various regions obtained from OpenStreetMap (https://www.openstreetmap.org/copyright, CC BY-SA 2.0) were utilized as data sources for plotting the map. Map data from © OpenStreetMap. https://www.openstreetmap.org/copyright. Color Gradient: Smooth transition from light blue (low density) to deep blue (high density), reflecting increasing intensity of cases per square kilometers.

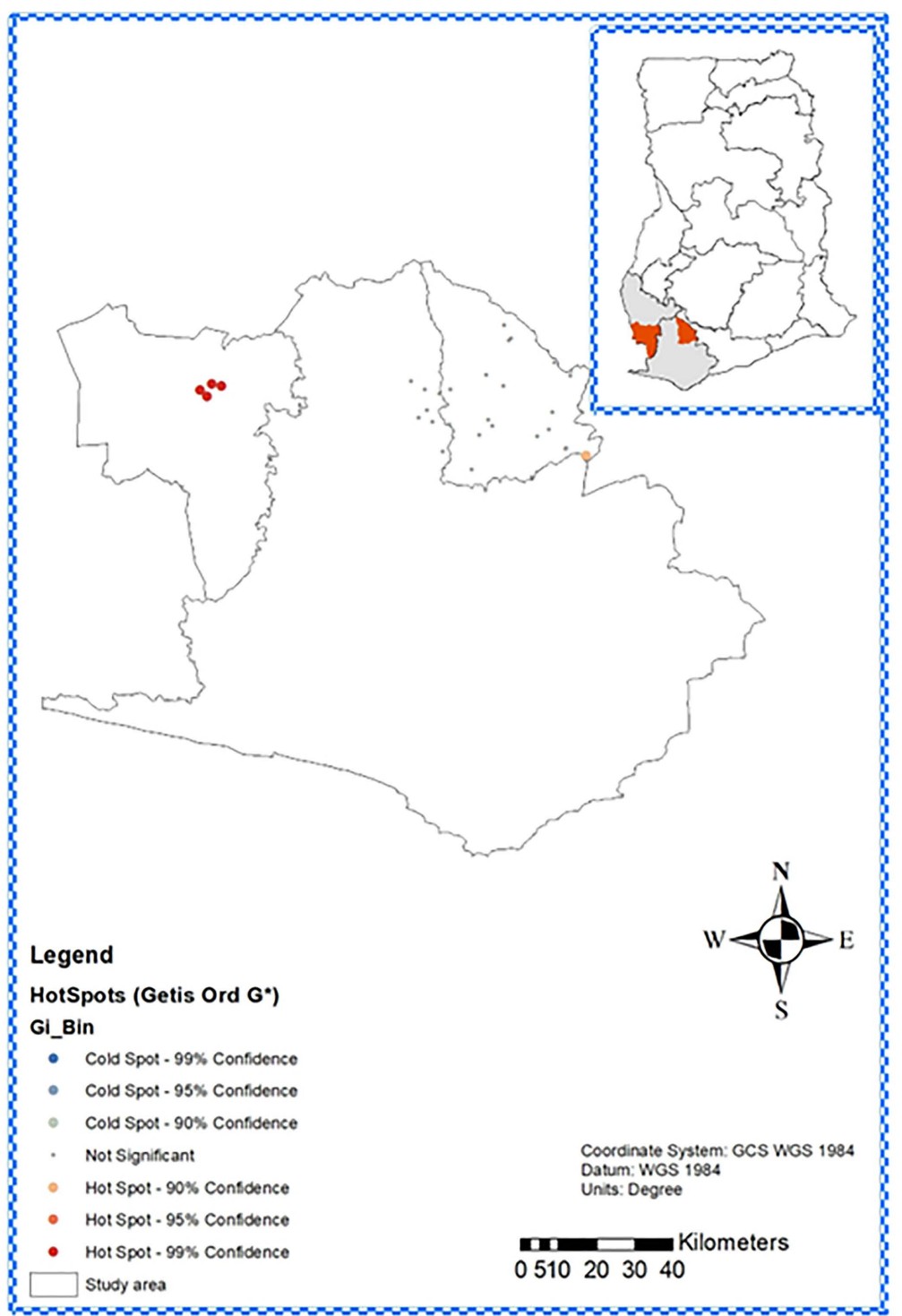

**Fig 6. Hotspots and coldspots of yaws prevalence in Wassa Amenfi East and Aowin Districts, Ghana.** Map was generated using ArcGIS 10.7.1 (Esri Inc., Redlands, California, USA). The shapefiles for Ghana and the various regions obtained from OpenStreetMap (https://www.openstreetmap.org/copyright, CC BY-SA 2.0) were utilized as data sources for plotting the map. Map data from © OpenStreetMap. https://www.openstreetmap.org/copyright. Light Blue- Cool spots, representing areas with low activity or sparse data points. Orange/Yellow: Moderate hotspots, indicating areas with intermediate activity or data point concentration. Dark Red: Hotspots with the highest intensity or concentration of activity. All at 90, 95 and 99% Confidence.

unsampled areas near Aowin's hotspot, estimating a prevalence range of 0.04% to 4.66% (Fig 7A). Empirical Bayesian kriging accounting for local variance produced similar estimated prevalence rates, 0.04% to 4.66% reinforcing the reliability of the predicted prevalence in the study area (Fig 7B).

## Discussion

This is the first study to provide critical insights into the prevalence and seroprevalence of yaws in Aowin and Wassa Amenfi East districts of Ghana, revealing a substantial disease burden, as well as the spatial distribution and predictive modeling of the disease. A large number of individuals across 37 basic schools in the two districts were screened, yielding an overall yaws prevalence of 0.63% (Aowin 3.85%, Wassa Amenfi East 0.31%). DPP testing confirmed 73 cases

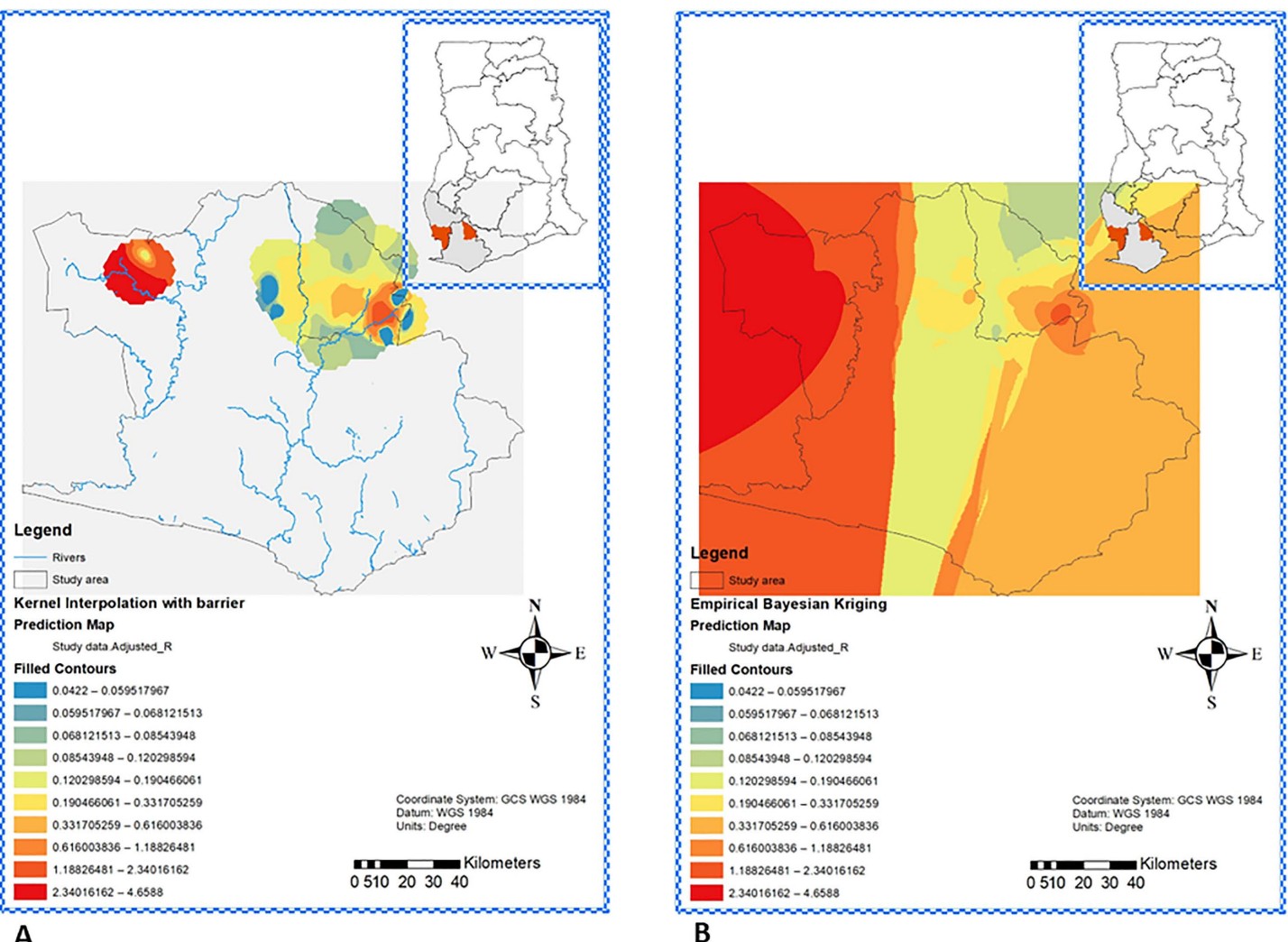

**Fig 7. Prediction of Yaws Prevalence in Unsampled areas in Wassa Amenfi East and Aowin Districts, Ghana.** Kernel Interpolation with Barrier Prediction of yaws prevalence in unsampled areas **(A)**. Colour from blue to red indicates lowest through to highest predicted prevalence areas. Empirical Bayesian Kriging Prediction of yaws prevalence in unsampled areas in Wassa Amenfi East and Aowin Districts, Ghana **(B)**. Colour from blue to red reflects the strength of Bayesian predictive probability. Maps were generated using ArcGIS 10.7.1 (Esri Inc., Redlands, California, USA). The shapefiles for Ghana and the various regions obtained from OpenStreetMap ([https://www.openstreetmap.org/copyright](https://www.openstreetmap.org/copyright), CC BY-SA 2.0) were utilized as data sources for plotting the maps. Map data from © OpenStreetMap. [https://www.openstreetmap.org/copyright](https://www.openstreetmap.org/copyright).

[https://doi.org/10.1371/journal.pntd.0013632.g007](https://doi.org/10.1371/journal.pntd.0013632.g007)

accounting for a seroprevalence of 62.39% (Aowin, 70.69% and Wassa Amenfi East, 54.24%). The active yaws prevalence was 0.62%, with significant district-specific variations (3.85% in Aowin, 0.29% in Wassa Amenfi East). Latent cases were identified in 0.02% of individuals from the Wassa Amenfi East district.

These findings align with previous studies reporting yaws as a persistent public health challenge in rural, tropical regions of Ghana [39,40] and other endemic regions of the world [41] though certain regions have documented notably low prevalence rates [42]. The observed DPP seroprevalence disparity between Aowin (70.69%) and Wassa Amenfi East districts (54.24%) may reflect differences in historic transmission patterns, surveillance efforts, and treatment interventions. It is noteworthy that the Wassa Amenfi East district conducted one round of total community treatment (TCT) with an 83.6% population coverage in 9 endemic communities, and one round of TTT was conducted in 31 endemic communities in 2022; this intervention may have contributed to the observed differences in our study. Although the study did not analyze the specific environmental or socioeconomic covariates to explain the differences, the variation in disease burden between the two districts could be influenced by rainfall patterns, humidity, sanitation conditions, and overcrowding [29,43]. Reports from the Ghana Statistical Service (GSS) estimates approximately 37.1% and 35.5% of Aowin and Wassa Amenfi East's population as multidimensionally poor, with the high deprivations experienced in toilet facilities, housing, overcrowding, lack of health insurance coverage, and portable water source [44,45] In recent years, these have been further compromised by activities of illegal miners and other forms of human activities that hamper sanitation [21,22]. These known risk factors could potentially account for the observed high prevalence and seroprevalence.

The prevalence rate recorded for children 1–5 years in the study population was 0.18% (Aowin 0.95% and Wassa Amenfi East 0%). Similarly, Wassa Amenfi East did not record seroprevalence rates for this age group, while Aowin recorded an RDT seroprevalence of 10.53% and a DPP seroprevalence of 100%. These findings in Aowin suggest ongoing transmission in the population and would require MDA intervention to achieve interruption of transmission [18].

Additionally, the 0.02% latent yaws prevalence in Wassa Amenfi East although lower than rates reported in other endemic regions like the Philippines (0.30%) [41] and other areas (2·45% to 31·05%) [40] remains of clinical and epidemiological significance as it can be a source of ongoing transmission later affecting the eradication of the disease. In contrast, a study in other Ghanaian districts that previously received azithromycin MDA for trachoma control reported no latent yaws cases (0%) in the screened population [46]. However, as this study did not specifically include household contacts or older community members, this may not fully represent the true burden of latent yaws in the area, as some cases could have been missed.

The clinical presentation of yaws in this study, characterized predominantly by ulcers (65.8%) and squamous macules (16.4%), aligns with previous findings from endemic regions, such as the Western Pacific, where ulcers were reported to be the predominant clinical form, marking them as a hallmark of primary and secondary yaws [43,47–51].

Aowin district had 82.9% of clinical forms presented being primary lesions compared to Wassa Amenfi East, 43.8%, while Wassa Amenfi East presented more secondary lesions (50.0%) than Aowin (24.4%). This pattern suggests more advanced disease stages in Wassa Amenfi East, which may be due to incomplete MDA activities, delayed treatment, leading to progression to multiple secondary stage lesions, or differences in healthcare and health-seeking access and behaviors. If not treated, these cases may progress to latent stages, potentially serving as reservoirs for yaws re-emergence in the population. This study aligns with prior reports indicating that yaws eradication requires high coverage treatment campaigns exceeding 90%, targeting latent cases, alongside focused treatment of active cases and their contacts [29,52]. Conversely, the high prevalence of primary lesions in Aowin may reflect rapid ongoing transmission, needing early high coverage azithromycin MDA to interrupt transmission [53–55]. Although the study did not include screening of adults within this population, no evidence of tertiary infection was observed in any of the communities visited. This finding aligns with reports indicating a reduction in tertiary stage disease, potentially attributable to the efficacy of antibiotic treatment and overall enhancements in healthcare. [55,56].

The predominant anatomical location of the lesions was on the legs (45.2%) and soles (15.1%) affirms earlier studies highlighting lower limbs as primary sites, maybe due to frequent trauma and exposure in children [7,49,57–59]. The district-specific differences, with Aowin showing exclusive leg (51.2%) and knee (17.1%) involvement, as compared to Wassa Amenfi East showing more leg (37.5%) and soles (21.8%), may reflect variations in environmental exposures or cultural practices, such as footwear use or clothing. The absence of hand, face, and head lesions in Aowin, contrasted with their presence in Wassa Amenfi East, warrants further investigation into potential behavioural or ecological factors.

Sex-based analysis showed a higher prevalence of yaws among males than females across the districts. These results align with previous studies suggesting that males are more likely to contract yaws. [11,60]. This may reflect behavioural differences, such as more outdoor activities, rough play, or occupational exposures in males, which can cause skin abrasions that allow bacteria to enter. The age distribution of yaws cases shows the highest prevalence among 11–15-year-olds (0.79% overall, 5.66% in Aowin, 0.38% in Wassa Amenfi East), suggesting that older children, especially those aged 10–15, are at greater risk- possibly due to increased outdoor play and less supervision, leading to skin injuries that facilitate transmission. The lower prevalence in children aged ≤ 5 years (0.19% overall, 0.96% in Aowin, 0% in Wassa Amenfi East) may indicate less exposure or effective early diagnosis and treatment through community health programs in Wassa Amenfi East district. [61]. Yet, the finding is of significant public health concern, as the presence of cases in this age group serves as a key indicator of ongoing transmission [18].

The spatial patterns observed in this study show that yaws is distributed differently in Wassa Amenfi East and Aowin districts. The High-High clusters and their hotspot observed in Aowin district generally reflect the transmission pattern and its environmental and socioeconomic drivers in these forest areas [29,42,61,62]. The high proportion (75.6%) of primary lesions and high density in the Aowin district suggest rapid ongoing transmission, and warrants early, high coverage (>90%) MDA to interrupt spread [29]. On the other hand, Wassa Amenfi East's Low-High cluster and 90% confidence hotspot observed in the southeastern area suggest lower transmission patterns possibly due to historic MDA interventions. Although the 50% secondary lesions and 0.02% latent cases identified indicate delayed treatment or incomplete MDA coverage, allowing disease progression [61,63] and TCT should be implemented not to miss any latent infections that could resurge later.

While our study examined a large number (11505 persons) of individuals in the two districts, our study has potential limitations. Cross reactivity between syphilis and yaws makes it difficult to differentiate between the two diseases based on serology alone. This lack of specificity, along with the potential for both diseases to coexist in the same areas, complicates epidemiological surveillance and diagnosis and a reliance on serologic tests alone could result in an overestimation of prevalence. In this study, we combined clinical examination findings and serology to improve diagnostic accuracy. While molecular tests such as polymerase chain reaction (PCR) offer higher specificity, they were deemed impractical in this field survey. The study was a school-based screening and did not include the adult population in the communities. Hence, this may not give a broader perspective of the prevalence and particularly the RDT seroprevalence rates for more targeted intervention. Further and wider serosurveys taking into consideration the parameters required for serosurveys, including serological testing for yaws latency [18] are needed to adequately investigate this area for a more targeted intervention and use of resources, taking particular attention to children 1–5 years, as they are indicators of interruption of transmission. Due to limited availability of serological test kits, only clinically suspected participants and individuals reported to have had untreated but resolved yaws-like lesions were tested. The estimates may not fully represent the true seroprevalence in the general population and may be biased toward higher seropositivity missing other possible latent infections. Nevertheless, focusing on clinically suspected individuals provides important insights into the serological profile of those most likely to be actively transmitting the disease, and is valuable for guiding targeted surveillance and intervention in endemic settings where universal testing is not feasible.

## Conclusion

This study found an overall yaws prevalence of 0.63% and a non-random, clustered distribution with notable hotspots of yaws in some parts of the districts. Kriging predictions suggest that risk persists in unsurveyed zones. Our study highlights the importance of serosurveys and geospatial modeling in gathering key data to guide surveillance, education, and MDA strategies in endemic regions. Overall, this research lays the groundwork for expanded serosurveys, contributing to the WHO's 2030 yaws eradication goal.

## Supporting information

**S1 File. Global Moran's I Autocorrelation report.** *Maps were generated using ArcGIS 10.7.1 (Esri Inc., Redlands, California, USA). The shapefiles for Ghana and the various regions obtained from OpenStreetMap (*https://www.openstreetmap.org/copyright*, CC BY-SA 2.0) were utilized as data sources for plotting the maps. Map data from © OpenStreetMap.* https://www.openstreetmap.org/copyright.
(PDF)

**S2 File. Local Indicator of Spatial Association, [LISA (High -Low clustering report)].** Maps were generated using ArcGIS 10.7.1 (Esri Inc., Redlands, California, USA). The shapefiles for Ghana and the various regions obtained from OpenStreetMap (https://www.openstreetmap.org/copyright, CC BY-SA 2.0) were utilized as data sources for plotting the maps. Map data from © OpenStreetMap. https://www.openstreetmap.org/copyright.'.
(PDF)

**S3 File. Nearest Neighbour summary.** Maps were generated using ArcGIS 10.7.1 (Esri Inc., Redlands, California, USA). The shapefiles for Ghana and the various regions obtained from OpenStreetMap (https://www.openstreetmap.org/copyright, CC BY-SA 2.0) were utilized as data sources for plotting the maps. Map data from © OpenStreetMap. https://www.openstreetmap.org/copyright.
(PDF)

## Author contributions

**Conceptualization:** Abigail Agbanyo, Alex Owusu-Ofori, Yaw Ampem Amoako, Richard Odame Phillips.

**Data curation:** Abigail Agbanyo, Michael Ntiamoah Oppong.

**Formal analysis:** Abigail Agbanyo, Michael Ntiamoah Oppong, Shadrach Mintah, Yaw Ampem Amoako.

**Investigation:** Abigail Agbanyo, Michael Ntiamoah Oppong, Ruth Dede Tuwor, Clement Tettey, Joseph Azabire, Owusu Boakye Yiadom, Yaw Ampem Amoako.

**Methodology:** Abigail Agbanyo, Alex Owusu-Ofori, Yaw Ampem Amoako, Richard Odame Phillips.

**Project administration:** Ruth Dede Tuwor, Yaw Ampem Amoako, Richard Odame Phillips.

**Resources:** Victor Yaw Morgan, Dennis Odai Laryea, Yaw Ampem Amoako, Richard Odame Phillips.

**Software:** Richard Odame Phillips.

**Supervision:** Alex Owusu-Ofori, Yaw Ampem Amoako, Richard Odame Phillips.

**Validation:** Ruth Dede Tuwor, Shadrach Mintah, Victor Yaw Morgan, Clement Tettey, Joseph Azabire, Owusu Boakye Yiadom, Dennis Odai Laryea, Alex Owusu-Ofori, Yaw Ampem Amoako.

**Visualization:** Michael Ntiamoah Oppong, Ruth Dede Tuwor, Shadrach Mintah, Victor Yaw Morgan, Clement Tettey, Joseph Azabire, Owusu Boakye Yiadom, Dennis Odai Laryea, Alex Owusu-Ofori.

**Writing – original draft:** Abigail Agbanyo, Yaw Ampem Amoako.

**Writing – review & editing:** Abigail Agbanyo, Michael Ntiamoah Oppong, Ruth Dede Tuwor, Shadrach Mintah, Victor Yaw Morgan, Clement Tettey, Joseph Azabire, Owusu Boakye Yiadom, Dennis Odai Laryea, Alex Owusu-Ofori, Yaw Ampem Amoako, Richard Odame Phillips.

## Acknowledgments

The authors extend their heartfelt thanks to the District Directors of Health Services and Education Service, Disease Control Officers, Community Leaders, Teachers, and pupils in Wassa Amenfi East and Aowin Districts for their valuable collaboration. We also express appreciation to Mr. Maxwell Adoko and the Skin NTDs Research group at KCCR for their support. Additionally, we are grateful to the WHO for supplying the DPP test kits.

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
